# Post-stress changes in the gut microbiome composition in rats with different levels of nervous system excitability

**Alla Shevchenko[1], Irina Shalaginova[1]\*, Dmitriy Katserov[1], Ludmila Matskova[3], Natalia Shiryaeva[2], Natalia Dyuzhikova[2]**

**1** Educational and Scientific Cluster "Institute of Medicine and Life Sciences (MEDBIO)", Immanuel Kant Baltic Federal University, Kaliningrad, Russia, **2** Pavlov Institute of Physiology of the Russian Academy of Sciences, Saint-Petersburg, Russia, **3** Department of Microbiology, Tumor and Cell Biology, Karolinska Institutet, Stockholm, Sweden

\* shalaginova_i@mail.ru

## Abstract

The gut-brain axis is a critical communication system influencing the interactions between the gastrointestinal tract (GI) and the central nervous system (CNS). The gut microbiota plays a significant role in this axis, affecting the development and function of the nervous system. Stress-induced psychopathologies, such as depression and anxiety, have been linked to the gut microbiota, but underlying mechanisms and genetic susceptibility remain unclear. In this study, we examined stress-induced changes in the gut microbiome composition in two rat strains with different levels of nervous system excitability: high threshold (HT strain) and low threshold (LT strain). Rats were exposed to long-term emotional and painful stress using the Hecht protocol, and fecal samples were collected at multiple time points before and after stress exposure. Using 16S rRNA amplicon sequencing, we assessed the qualitative and quantitative changes in the gut microbiota. Our results revealed distinct microbial diversity between the two rat strains, with the HT strain displaying higher diversity compared to the LT strain. Notably, under prolonged stress, the HT strain showed an increase in relative abundance of microorganisms from the genera *Faecalibacterium* and *Prevotella* in fecal samples. Additionally, both strains exhibited a decrease in *Lactobacillus* abundance following stress exposure. Our findings provide valuable insights into the impact of hereditary nervous system excitability on the gut microbiome composition under stress conditions. Understanding the gut-brain interactions in response to stress may open new avenues for comprehending stress-related psychopathologies and developing potential therapeutic interventions targeted at the gut microbiota. However, further research is needed to elucidate the exact mechanisms underlying these changes and their implications for stress-induced disorders. Overall, this study contributes to the growing body of knowledge on the gut-brain axis and its significance in stress-related neurobiology.

**Data Availability Statement:** All data files are available from the https://www.ncbi.nlm.nih.gov/bioproject/PRJNA831893.

**Funding:** This research was funded by the Russian Federal Academic Leadership Program Priority 2030 at the Immanuel Kant Baltic Federal University (IKBFU) - Irina Shalaginova, Dmitriy Katserov; Natalia Shiryaeva, Natalia Dyuzhikova by State Program 47 SP «Scientific and technological development of the RF», topic 0134-2019-0002 (Pavlov Institute of Physiology, Russian Academy of Sciences).

**Competing interests:** The authors have declared that no competing interests exist.

## Introduction

The gastrointestinal tract (GI) and the central nervous system (CNS) interact via the bidirectional biochemical signaling between neurons of the CNS and the GI, the gut–brain axis. There are three main pathways by which the microbiota exerts its effect on the function of the nervous system: through modulation of the immune response; through the metabolism of hormones, neuropeptides and neurotransmitters, and through direct influence on neurons and neural signaling [1]. The enteric nervous system and the vagus nerve provide one of the pathways through which the gut microbiota influences the brain. It has been shown that bacterial metabolites stimulate afferent neurons of the enteric nervous system [2]. Recent research identified sensory gut cells called "neuropod cells" that can form connections with the vagus nerve. These connections enable rapid information transfer from the gut to the brain [3]. The latest data indicate a significant role of the gut microbiota in the dysfunction of the Hypothalamic-Pituitary-Adrenal axis (HPA axis) and, consequently, in neuroendocrine dysregulation [4].

The gut microbiota has recently been identified as a factor that actively influences the functional state of the brain and host behavior. In particular, the gut microbiota plays a role in the development of neuroinflammation, which in recent years has been considered as one of the factors in the pathogenesis of post-stress psychopathologies (such as depression, anxiety disorders). The molecular mechanisms of microbiota-modulated neuroinflammation, acting both locally and systemically, have begun to be elucidated. For example, gut microbiota metabolites have been shown to inhibit host histone deacetylases (HDACs), which, as key modifiers of histones, control the assembly of transcriptional complexes, facilitating the active state of the host genome [5, 6]. Persistent changes in the intestinal microbiota can lead to increased production of proinflammatory cytokines, causing increased intestinal permeability [7]. This can mediate increased transfer of bacterial lipopolysaccharides and neuroactive metabolites into the bloodstream, and eventually into the brain, causing neuroinflammation in mammals [8].

Intestinal dysbiosis may also trigger immune system stimulation, which, combined with increased intestinal barrier permeability, may lead to local and systemic inflammation along with enteric neuroglial activation, which may lead to alpha-synuclein pathology [9].

Stress alters the composition of the gut microbiota in mammals, in particular by decreasing the relative abundance of *Lactobacillus* and *Bifidobacterium* [10; 11] and increasing *Odoribacter* and *Mucisprillum* [12] and *Clostridium* [13]. For example, knowledge of the exact taxonomic changes of the microbial community under the influence of stress, can provide useful hints for the choice of appropriate prebiotics and probiotics.

Among the individual risk factors for post-stress disorders, the hereditary excitability of the nervous system is of particular interest. Two strains of rats with different levels of nervous system excitability were obtained as a result of long-term selection [14]. These strains were shown to differ in their behavioral responses to stress, and their nervous tissues were found to differ at the cellular and molecular levels in the responses during post-stress recovery. In rats with low excitability (high threshold, strain HT), a prolonged treatment with a stressor induced depressive-like behavioral symptoms, whereas rats with high excitability (low threshold, strain LT), showed the emergence and maintenance of typical compulsive movements [14]. Previously, we showed that the level of excitability of the nervous system also affects the severity and dynamics of post-stress inflammation in both blood and brain: in response to long-term stress, the neutrophil/leukocyte ratio and the number of microglial cells in the hippocampus are increased in highly excitable animals [15].

The goal of this study is to investigate alterations in the gut microbiota of rat strains with high and low excitability both under normal conditions and following chronic stress exposure. Given that these rat strains exhibit varying susceptibility to stress in terms of post-stress

neuroinflammation characteristics and behavioral disturbances, understanding the precise taxonomic changes in the microbial community in response to stressors will provide insights into the influence of individual stress susceptibility on the microbiota, with broader implications for personalized medicine and interventions in stress-related disorders.

## Materials and methods

### Animals and stress protocol

Experiments were performed on 5-month-old adult male rats of two strains with different levels of the peripheral and central nervous system excitability [14]. Rats were selected for 80 generations with respect to high threshold (HT strain) or low threshold (LT strain) excitability of the tibial nerve (n. tibialis) to electric current, low excitable and highly excitable, respectively. The strains are included in the biocollection of the Pavlov Institute of Physiology of the Russian Academy of Sciences (No. GZ 0134-2018-0003), patents for breeding inventions No. 10769 and 10768 issued by the State Commission of the Russian Federation for Examination and Protection of Breeding Inventions, registered in the State Register of Protected Breeding Inventions on January 15, 2020.

The excitability thresholds of the of the tibial nerve (n. tibialis) in rat strains were determined based on the motor response of the hind limb when an active electrode was inserted into the gastrocnemius muscle, and single rectangular pulses with a duration of 2 milliseconds were applied.

All animals were maintained under standard vivarium conditions. The diet for the laboratory rats consisted of commercially available pelleted feed designed to provide the necessary nutrients. Protein: approximately 18–24% of the diet; carbohydrates: 40–70%; fats: 4–9%. The diet included supplements of vitamins and minerals. Laboratory rats had continuous access to water. All animal experiments were performed in accordance with the guidelines of the Council of the European Community (86/609/EEC) on the use of animals for experimental purposes. The protocol was approved by the Commission for the Humane Treatment of Animals of the Pavlov Institute of Physiology RAS. Experimental animals were exposed to long-term emotional and painful stress according to the Hecht protocol [14]: every day for 15 days the animals were exposed to 6 unsupported (10 seconds each) and 6 current-enhanced (2.5 mA, 2 ms) light signals. According to the scheme, combinations of conditioned and unconditioned stimuli were not repeated, but alternated with a probability of 0.5, which did not allow the animals to develop a conditioned reflex.

In each group of HT and LT animals there were 12 rats, which were divided into 6 animals in two subgroups–experimental and control. To study the dynamics of changes in the composition of the gastrointestinal microbiota, samples of biological material (feces) were collected at the following time intervals–(A) before stress exposure, (B) seven days later and (C) 24 days after the end of the stressor protocol. Feces from control animals were collected at the same time points as the experimental animals. Sampling was performed in special cages, in which each animal was restrained, to prevent coprophagia. Collected samples were stored at -70C without preservative agents.

## DNA extraction from fecal samples and 16S rRNA amplicon sequencing

Total DNA extraction was conducted in accordance to protocol «Q INRA» of International Human Microbiome Standards consortium with minor modifications [16, 17]. After slow thawing at +4˚C, from 150 to 200 mg of fecal matter from each sample were distributed

among 2 ml shatter-resistant tubes (#2641-0B, Lodi, CA, USA) prefilled with 3:1 mix of 0.1 mm (#G8772) / 0.5 mm (#G8893) glass beads (Merck KGaA, Germany), which were later subjected to bead-beating in Minilys personal homogenizer (#P000673-MLYS0-A.0, Bertin technologies, France) for 180s at 3000 RPM after the addition of ASL lysis buffer (#19082, QIAGEN, Hilden, Germany). Such protocol steps as partial elimination of indigested proteins with 10M ammonium acetate, isopropanol nucleic acid precipitation, RNAse treatment and silica column purification of DNA extract with the QIAamp DNA Stool Mini kit (#51504, QIAGEN Hilden, Germany), were carried out without any changes. DNA integrity was evaluated by electrophoresis of the extract in 1% agarose gel. All nucleic acid concentration measurements were conducted using Qubit 2.0 fluorometer (#Q32866, Thermo Fisher Scientific, Waltham, MA, USA) with proprietary Qubit dsDNA HS Assay Kit (#Q32851, Thermo Fisher Scientific, Waltham, MA, USA).

Amplicon library preparation included single PCR reaction with degenerate 16S rRNA gene V4 hypervariable region primers F515 (5′–GTGBCAGCMGCCGCGGTAA–3′) and 806R (5'-GGACTACHVGGGTWTCTAAT-3'), modified to include linker sequence, 12 bp index and 0–7 bp "heterogeneity spacer" [18]. Amplification was performed using the qPCRmix-HS SYBR reaction mix (Evrogen, Russia) on CFX96 real-time PCR system (BioRad, Hercules, CA, USA), with the following amplification program: 95˚C/3 min of initial denaturation followed by 28 cycles of 95˚C/20 s denaturation, 58˚C/20 s annealing, 72˚C/20 s elongation and final elongation at 72˚ for 30 seconds. PCR products quality and size was evaluated by electrophoresis in 1,5% agarose gel. Amplicons were purified using QIAquick Gel Extraction Kit (#28704, QIAGEN, Hilden, Germany) and their concentration measured prior to pooling at equimolar concentration. Final sample pool was purified and concentrated with AMPureXP magnetic beads (#A63881, Beckman Coulter, Brea, CA, USA). Paired-end sequencing was performed on the Illumina Miseq platform, using a MiSeq 500-cycle PE Reagent kit V2 with Nano flow cell (Illumina Inc., San Diego, CA, USA) according to manufacturer's instructions.

## Data analysis

The quality of the raw reads was checked using FastQC [19]. First, the reads were filtered using fastp-0.12.4 (total reads before filtering 1023074, after filtering 972205.) [20] and then demultiplexed using deML [21]. Demultiplexed reads were analyzed with the Quantitative In-sights Into Microbial Ecology 2 (Qiime2-2020.14) software pipeline [22]. Denoising and adapter trimming of sequence data was performed using q2-dada2 (denoise-paired) plugin [23]. In addition, the data was analyzed using Rstudio, R version 4.3.1 (2023-06-16) (phyloseq, ggplot2, tidyverse) [24].

The MAFFT algorithm [25] was used to align all amplicon sequence variants (ASVs), and the Fasttree algorithm [26] was used to generate a rooted phylogenetic tree. The taxonomy was annotated to the Greengenes database [27] using a naive Bayes classifier via the q2-feature-classifier plugin.

To estimate the alpha diversity within the microbial community, the Shannon and Chao1 indices were calculated, which describe the richness and evenness of each sample. QIIME2 taxa barplot algorithm was used to define the relative abundance of different levels of the taxonomy.

## Statistics

The distribution of the data was evaluated by the Shapiro-Wilk test using the GraphPad Prism software (v.6, GraphPad Software Inc.). Parametric (paired and unpaired t-test, ANOVA and

the Greenhouse-Geisser correction method) tests were also performed using GraphPad Prism software. All data are presented as mean and SEM.

## Results

### Interstrain differences in the threshold voltage levels in LT and HT rat strains

In order to validate the strains of rats with high and low excitability, Fig 1 presents the differences in the voltage threshold at which the motor reaction appeared.

### Microbiome α-diversity indexes in rat strain with high and low excitability

The absence of significant differences in microbiota diversity in rats of the control group at different time points during the experiment indicates in favor of the stability of these indices (Fig 2A). This fact made it possible to join the data of control animals into a general group when comparing the case/control (Fig 3).

In the gut microbiome of intact rats of the highly excitable strain (LT), there was a decrease in the α-diversity of the gut bacterial community compared to the low-excitable (HT) strain (the Shannon and Chao1 indexes were significantly reduced, Fig 2B).

We did not find any statistically significant effect of stress on alpha diversity indexes in LT and HT rats (Fig 3).

### Gut microbiota profile in rat strain with high and low excitability in normal condition and at different time points after stress

The stool microbiome of high threshold rat strain (HT—low excitable) that were not subjected to stress (intact) contained significantly fewer bacteria from the genera *Blautia*, *Collinsella*, *Methanobrevibacter*, and more bacteria of the genus *Prevotella* than the microbiome of rats with a low threshold (LT—high excitable) of excitability of the nervous system (Fig 4).

Since we evaluated the composition of the intestinal microbiota in the experimental groups before exposure to stress, it was possible to compare the corresponding control groups with experimental animals before the stress exposure. We found that in one case (Fig 5, g. *Prevotella*) the control HT rats significantly differed from the experimental group before stress, which makes it incorrect to compare control groups with experimental ones to assess the effect of stress in this case.

Statistical analysis of the dynamics of changes in the relative abundance of bacteria in the rat gut before and after the stress exposure showed a significant decrease in g. *Lactobacillus* by day 24 as compared to the pre-stress period in both rat strains (Fig 5). The relative abundance of g. *Prevotella*, g. *Faecabacterium* increased significantly in the gut microbiome of rats in response to stress only in the group of low-excitable HT animals.

Before stress in rats of both strains, the relative abundance of Firmicutes prevails over all others phylum (Fig 6). After stress (on the 7th day) in rats of the HT strain Firmicutes (31%) are displaced by Bacteroidetes (68%), but at the same time (7 days after stress) in animals of the LT strain Firmicutes still prevail over Bacteroidetes.

## Discussion

Earlier, we found an increased level of il1$\beta$ mRNA in the blood of intact animals of the highly excitable LT strain compared to the low excitable HT strain [28]. This fact is consistent with the reduced alpha diversity (Shannon, Chao1 indices) of the intestinal microbiome here in intact LT rats. The high alpha diversity of the gut microbiota is associated with lower levels of

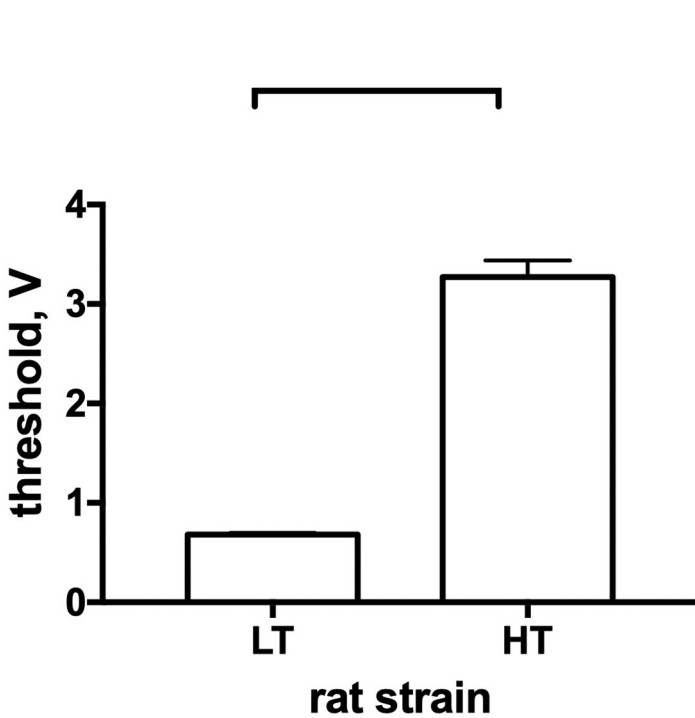

**Fig 1. Interstrain differences in the volage threshold in LT and HT rat strains.** HT—animals with high excitability threshold (n = 32), LT–animals with low excitability threshold (n = 34); the graphs represent the mean and SEM; **** p< .0001 (unpaired t-test).

pro-inflammatory cytokines in the blood as was shown in humans: people with high alpha diversity of the gut microbiota have lower levels of proinflammatory cytokines in the blood than people with low alpha diversity of the gut microbiota [29]. This has also been shown in an animal model [30], where the authors demonstrated an association between the increased levels of proinflammatory cytokines (IL1β and IL-18) in the rat brain when exposed to an inflammatory stimulus and changes in the gut microbiota. It was found that inflammation leads to a decrease in the index of microbiota diversity.

Stress-induced changes in CNS functions are associated with altered immunoregulatory responses, which can be significantly associated with changes in the microbiota [31], as demonstrated in male C57BL/6 mice that were exposed to chronic social damage stress. It has also been shown [32] that traumatic peripheral nerve injury (physical stress) can alter the microbial diversity and stability of the gut, as indicated by a decrease in Chao1 and Shannon indices in experimental animals.

Thus, there is an association between the alpha diversity of the gut microbiota and the levels of pro-inflammatory cytokines in the blood and brain in both animals and humans. However, the exact causal relationship between these factors has so far not been established.

The genetic characteristics of different rodent strains appear to influence the diversity of the gut microbiota. Mice of the C57BL/6J strain showed higher indices of diversity and stability of the intestinal microbiota under normal conditions compared to mice of the BALB/c strain. Interestingly, at the same time, BALB/c mice are considered more susceptible to

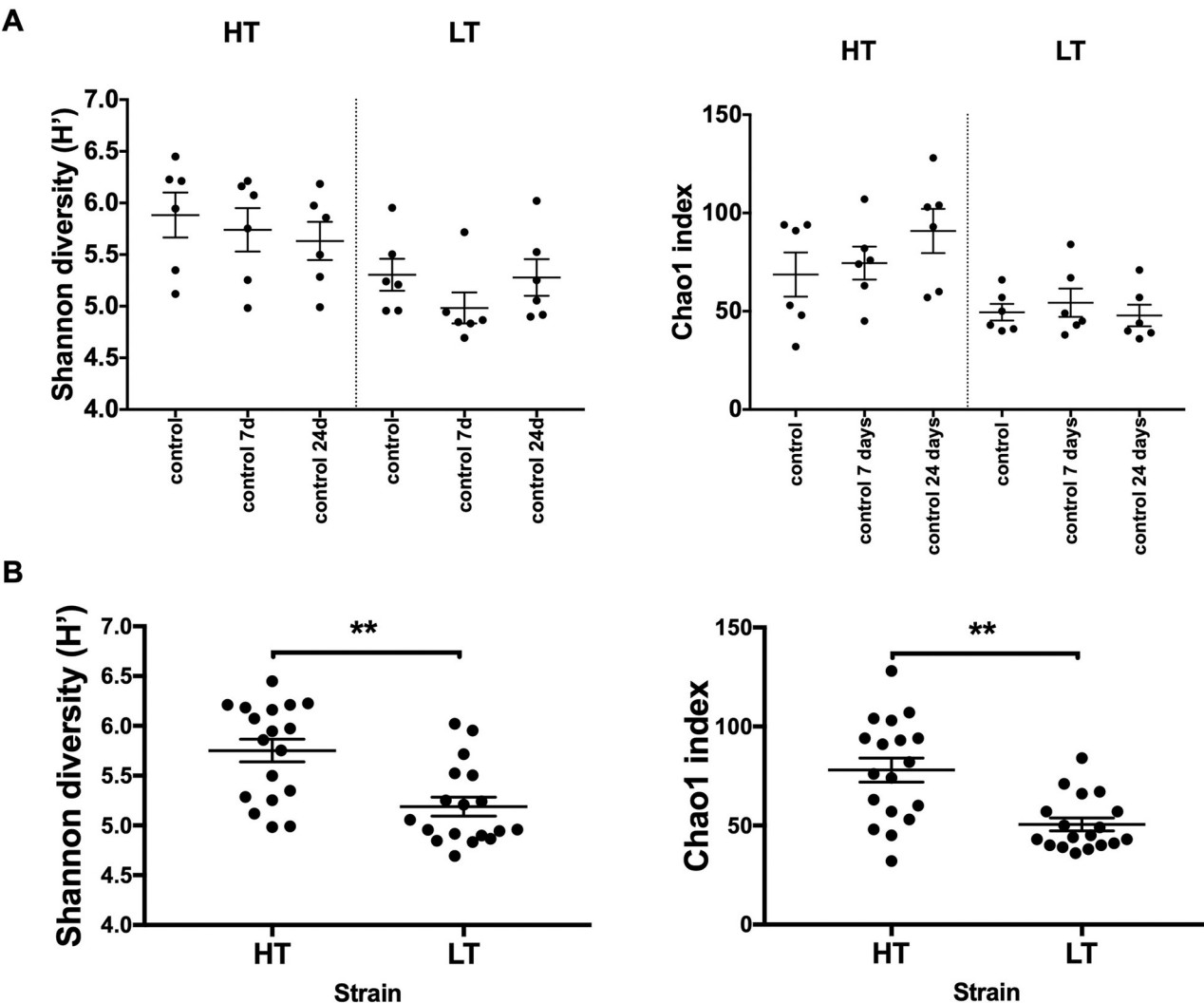

**Fig 2. Differences in alpha diversity indices in control groups of HT and LT rat strains.** The horizontal axis on panel A shows the animals of the control groups at the corresponding time after the end of stress in the experimental groups. The horizontal axis on panel B shows the intact animals of the consolidated control. HT–high threshold, low-excitable rats; LT–low threshold, high-excitable rats; the graphs represent the mean and SEM; ** $p < 0,01$, in A–paired t-test, in B—unpaired t-test.

immune dysfunction [33]. In response to stress, as well, they demonstrate higher levels of corticosterone and greater severity of anxiety-like symptoms compared to C57BL/6J mice [34].

Thus, we found a significant decrease in the diversity of the intestinal microbiota in rat strains with genetically determined high excitability compared with low-excitable rats, may contribute to a lower adaptability of the LT strain to stress in terms of post-stress inflammation and behavioral disorders [15]. However, it is necessary to conduct a study aimed at correcting the intestinal microbiota in highly excitable rats (for example, treatment with probiotics). Despite the recommendations to take into account possible dynamic processes in the microbiota composition of intact animals [35], in most studies, stool sampling for analysis is done in the control and experimental group once: for illustration [36, 37]. At the same time, there is evidence of significant changes in alpha diversity indexes in intact mature rodents kept under

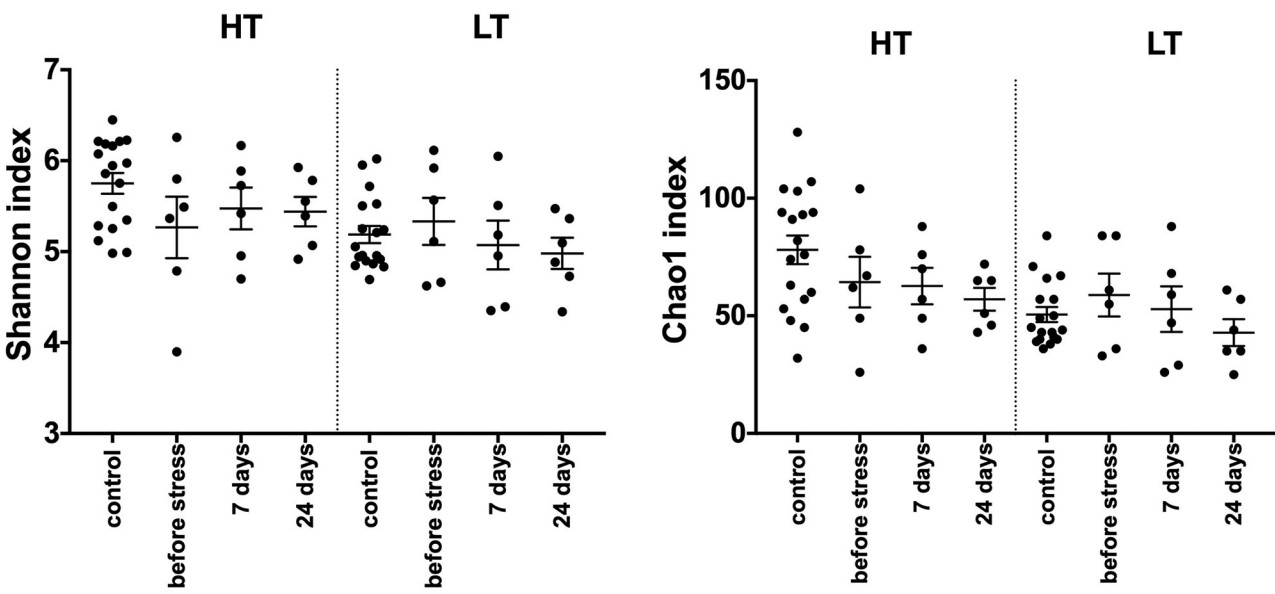

**Fig 3. Alpha diversity indices in control and experimental groups of HT and LT rats after stress.** The horizontal axis shows the animals of the control groups (n = 18 in HT strain and n = 18 in LT strain) and the experimental groups (n = 6 in each group) at the different time points after the end of stress. HT–high threshold, low-excitable rats; LT–low threshold, high-excitable rats; the graphs represent the mean and SEM; one-dimensional variance analysis with repeated measurements (ANOVA), using the Greenhouse-Geisser correction method; case/control—unpaired t-test p > 0,05.

standard conditions if biomaterial sampling is carried out every week [38]. Taking to account for these changes, we carried out sampling not only in animals of the control group at 3 time points, but also in the experimental group before stress. According to our results, the alpha diversity in the animals of the studied strains is stable and did not significantly change in the control throughout the experiment.

We have identified interstrain differences in the abundance of bacteria of some genera. The stool microbiome of intact low excitable rats (HT) contained significantly fewer bacteria from the genera *Blautia*, *Collinsella*, *Methanobrevibacter*, and more bacteria of the genus *Prevotella*

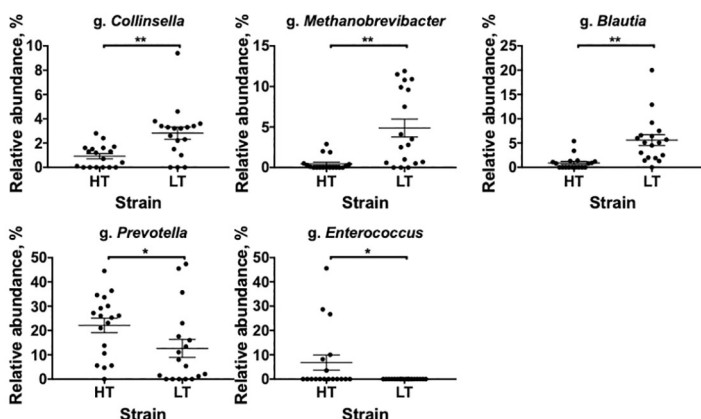

**Fig 4. Differences in the relative abundance of some bacterial genera in the gut microbiome of the HT and LT rat strains.** HT–high threshold, low-excitable intact rats; LT–low threshold, high-excitable intact rats; n = 18 in each group; the graphs represent the mean and SEM **p<0.01 *p<0.05, unpaired t-test.

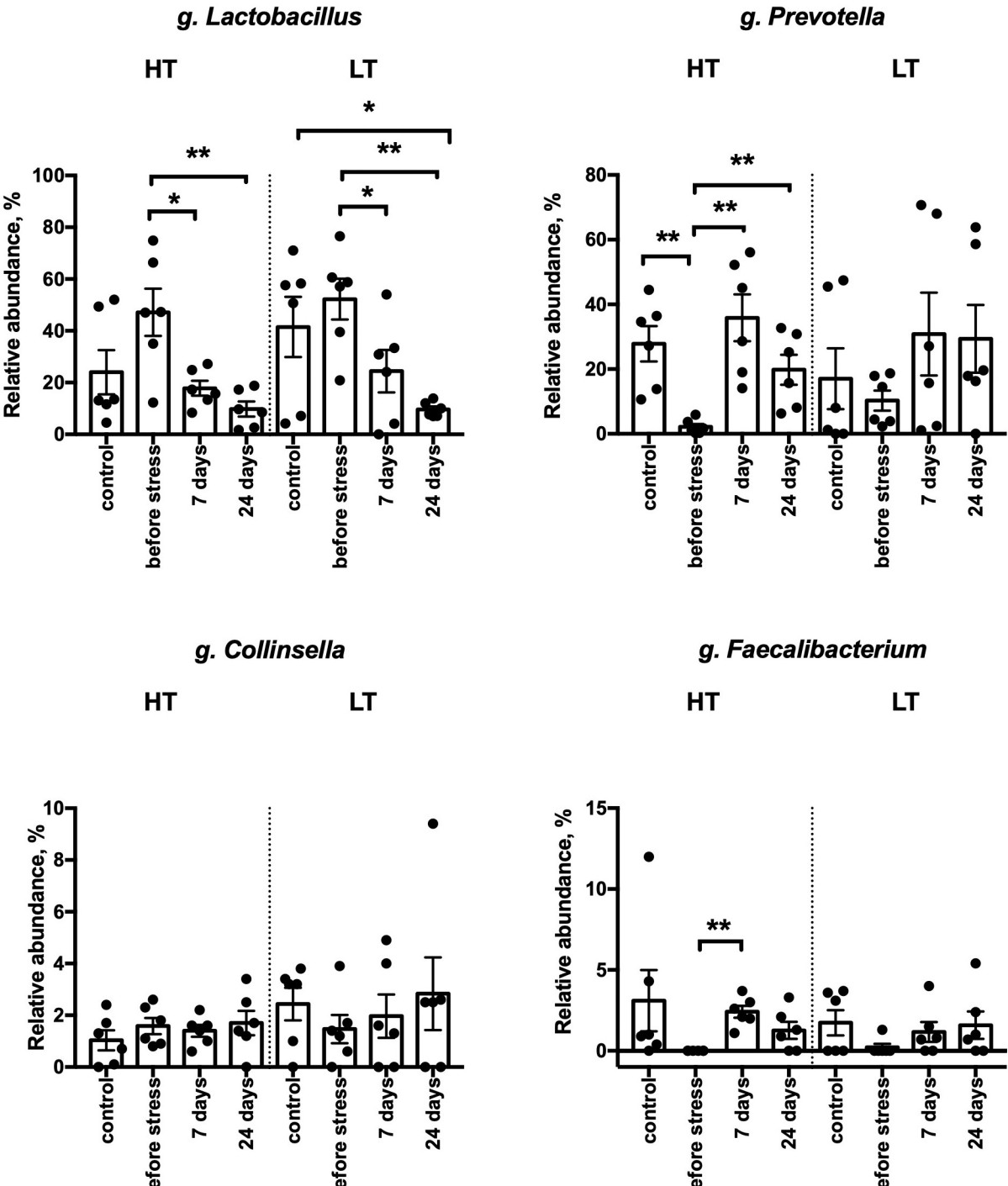

**Fig 5. Differences in the relative abundance of top bacterial genera in control and stress groups of HT and LT rat strains.** HT–high threshold, low-excitable rats; LT–low threshold, high-excitable rats; control–rats without stress exposure; before stress–experimental rats before stress exposure; 0, 7, 24 –days after the end of stress exposure in experimental groups; n = 6 in each group; the graphs represent the mean and SEM **p<0.01 *p<0.05, case/control—unpaired t-test; experimental rats in different time points—one-dimensional variance analysis with repeated measurements (ANOVA), using the Greenhouse-Geisser correction method and paired t-test.

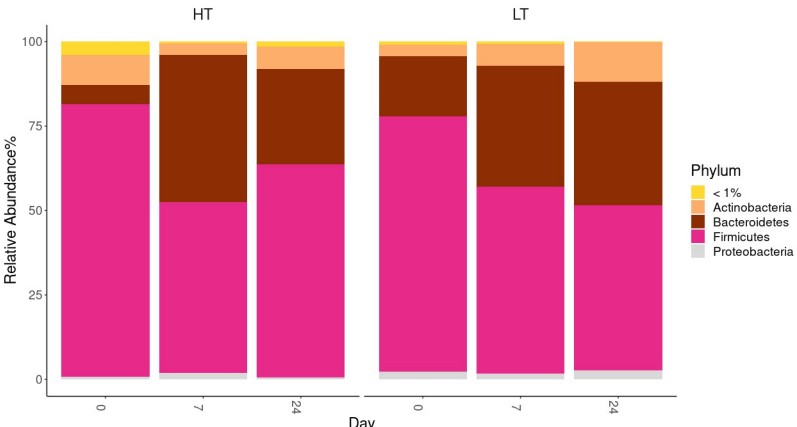

**Fig 6. Altered composition in the gut microbiota at phylum levels in stressed groups of HT and LT rat strains.**
HT–high threshold, low-excitable rats; LT–low threshold, high-excitable rats; 0, 7, 24 –days after the end of stress exposure in experimental groups.

than the microbiome of high excitable (LT) rats. But in the case of *g. Prevotella*, we found significant differences between the control group and intact animals of the experimental group, so the interstrain differences in the relative abundance of this genus of bacteria should be interpreted with caution. Nevertheless, our results show that after the stress, the microbiome of the stool of rats with a high threshold of excitability changes more dynamically. Since we controlled the factors that could affect the composition of the microbiota (diet, cage effect, etc.), it is possible that "spontaneous" changes in the *g. Prevotella* are associated with the genetic characteristics of the HT strain. To date, these rat strains have not been genotyped, which makes it difficult to find a connection between the microbiome and the genetic characteristics of the host. However, in animals with genetically determined contrast excitability, not only behavioral differences were revealed, it is known that highly excitable rats (LT) have increased thyroid and HPA axis activity (high basal corticosterone level), higher levels of dopamine in the amygdala, higher levels of calcium in the brain and calmodulin in the hippocampus compared to the HT strain [39].

We also found changes in the relative abundance of certain genera of gut bacteria in rat feces in response to stress. We observed a decrease in the relative abundance of *Lactobacillus* bacteria in animals of both strains at 24 days after exposure to a stressor, as compared to the pre-stress state. This is consistent with the published data that stressor exposure alters the composition of the bacterial community and decreases the relative abundance of the genus *Lactobacillus* [40]. The ability of *Lactobacillus* bacteria to suppress the expression of proinflammatory cytokines in stressed animals has recently been demonstrated [41]. It is possible that the post-stress decrease in these bacteria is associated with the increased expression of proinflammatory cytokines that we observed in rats of both strains, but which is more pronounced in the highly excitable LT animals. But supplementary research is needed to check whether the level of cytokines in the brain will change when the composition of the intestinal microbiota is restored.

We have shown that the relative abundance of microorganisms of the genus *Prevotella* increases in the intestines of rats after stress, but only in the group of low-excitable animals of the HT strain. It was previously shown that the use of high and low temperatures as stressors caused a significant increase in the relative abundance of the *Prevotellaceae* family in the

intestines of mice [42]. The genus *Prevotella* was significantly more abundant in the intestinal microflora of rats in stress-induced models of irritable bowel syndrome (IBS) [43].

We also observed an increase in the relative abundance of bacteria of the genus *Faecalibacterium* in the stool of low-excitable rats (HT) on the 7th day after stress, as compared to the state before stress exposure. There is evidence that representatives of the genus *Faecalibacterium* have a preventive and therapeutic effect on chronic depressive and anxiety-like conditions caused by stress in rats [44]. It is possible that such an increase in the occurrence of bacteria of the genus *Faecalibacterium* may be one of the factors that increase the adaptive potential of the HT strain and explain the absence of post-stress behavioral abnormalities in HT animals up to 24 days, as well as their lower severity post-stress neuroinflammation in comparison to the more susceptible LT strain [15]. Nevertheless, by the 24th day after stress, the relative abundance of *Faecalibacterium* bacteria in the feces of HT animals returned to the baseline levels.

We also found a change in the ratio of microorganisms belonging to the *Firmicutes* and *Bacteroidetes* types in the gut microbiota in response to long-term emotional and pain-induced stress in rats of both strains. Prior to stress, the relative abundance of the *Firmicutes* members pre-vails over all others in rats of both strains, which is consistent with the published data on the normal intestinal microbiota of both rodents and humans [45, 46]. After stress (on day 7) in HT animals, *Firmicutes* are displaced by *Bacteroidetes*, but at the same time (7 days after stress) in LT animals *Firmicutes* still outnumber *Bacteroidetes*. These results are consistent with previously published studies showing that stress can lead to an imbalance in the gut microbiota and a change in the ratio between *Firmicutes* and *Bacteroidetes*. For example, animal studies have shown that stress can increase the number of *Bacteroidetes* bacteria and decrease the representation of the *Firmicutes* in the intestinal microbiota [13].

Thus, the obtained results suggest that the intestinal microbiota of the HT strain is normally more diverse and less susceptible to changes over time. However, under the influence of a stressor, changes in the composition of the microbiome are more dynamic than in animals with high excitability of the nervous system (LT). Nevertheless, as our previous results show, the HT strain is more resistant to stress, since behavioral disorders appear later after stress exposure and signs of neuroinflammation are less pronounced compared to the LT strain. It can be suggested that the initial diversity of the gut microbiota may compensate for transient post-stress fluctuations in the relative abundance of individual genera of the gut microbiota. However, this assumption requires additional verification in experiments with manipulation of the gut microbiota composition in high and low-excitable rat strains.

## Conclusion

Thus, in rats with high and low hereditary excitability of the nervous system, a different composition of the microbiota and its specific changes under the influence of stress were revealed.

Greater diversity in the microbiota composition may be associated with greater resistance to stress in rats of the low excitable LT strain. Knowing about the effect of stress on the intestinal microbiota of both highly excitable and low-excitable animals, and the possibilities of symbiont microorganisms to modulate the functional state of the central nervous system, it is necessary to check whether targeted changes in the composition of the microbial community or the content of microbial metabolites will have an anti-inflammatory effect in the nervous tissue and whether such intervention is capable of normalizing animal behavior. The metabolism of representatives of the bacterial genera *Faecalibacterium*, *Prevotella* and *Lactobacillus* deserves a detailed study at the molecular level. As our understanding of the mammalian microbiome deepens, monitoring and identifying modifiable features that may contribute to stress tolerance will have important clinical implications in turbulent times.

## Supporting information

**S1 Data. HT and LT excitability thresholds.**
(PZFX)

**S2 Data. HT and LT Shannon-Chao1 indexes.**
(PZFX)

**S3 Data. HT and LT relative abundance of microbiota (genius level).**
(PZFX)

## Author Contributions

**Conceptualization:** Irina Shalaginova, Natalia Dyuzhikova.

**Formal analysis:** Alla Shevchenko, Irina Shalaginova.

**Funding acquisition:** Irina Shalaginova, Ludmila Matskova, Natalia Dyuzhikova.

**Investigation:** Alla Shevchenko, Irina Shalaginova, Dmitriy Katserov, Natalia Shiryaeva.

**Methodology:** Natalia Dyuzhikova.

**Project administration:** Irina Shalaginova.

**Supervision:** Ludmila Matskova, Natalia Dyuzhikova.

**Visualization:** Alla Shevchenko.

**Writing – original draft:** Irina Shalaginova.

**Writing – review & editing:** Ludmila Matskova, Natalia Dyuzhikova.

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
