## [Decision Letter · Decision Letter 0]

6 Oct 2023

PONE-D-23-27857Post-stress changes in the gut microbiome composition in rats with different levels of nervous system excitabilityPLOS ONE

Dear Dr. Shalaginova,

Thank you for submitting your manuscript to PLOS ONE. After careful consideration, we feel that it has merit but does not fully meet PLOS ONE’s publication criteria as it currently stands. Therefore, we invite you to submit a revised version of the manuscript that addresses the points raised during the review process.

 In addition to noted major concerns regarding the scientific setup, data presentation and interpretation, both reviewers noted serious concerns regarding the organization, presentation, and clarity of the manuscript. The authors must pay more attention to the quality of the work and presentation in a majorly revised manuscript.

We look forward to receiving your revised manuscript.

Kind regards,

Brenda A Wilson, Ph.D.

Academic Editor

PLOS ONE

Journal Requirements:

When you resubmit, please ensure that you provide the correct grant numbers for the awards you received for your study in the ‘Funding Information’ section."

Additional Editor Comments:

This manuscript barely made it past the initial editorial stage of review. The content, message, data analysis and interpretation, and figures were very poorly prepared, and the manuscript requires major rewriting.

Reviewers' comments:

Reviewer's Responses to Questions

**Comments to the Author**

1. Is the manuscript technically sound, and do the data support the conclusions?

Reviewer #1: No

Reviewer #2: Yes

2. Has the statistical analysis been performed appropriately and rigorously? 

Reviewer #1: No

Reviewer #2: Yes

3. Have the authors made all data underlying the findings in their manuscript fully available?

Reviewer #1: No

Reviewer #2: Yes

4. Is the manuscript presented in an intelligible fashion and written in standard English?

Reviewer #1: No

Reviewer #2: Yes

5. Review Comments to the Author

Reviewer #1: The article needs a major revision and language editing before it could be considered for publication. The manuscript’s content is also rudimentary and structured very poorly. See comments below:

Major comments:

1. Introduction: The three main pathways mentioned should also include the HPA axis and the enteric nervous system.

2. Objective: What is the vision behind the experiments performed? What is the conceptual goal and what are the implications of this study that warrants the reader’s attention?

3. Methods: For all software used, please include their respective versions. Even with published data regarding the HT or LT strains, the authors should consider including validation of the strains for clarity of the manuscript. As this is a microbiome study, content of diet should also be mentioned. Preprocessing step is crucial and not mentioned. For reference:

a. https://www.nature.com/articles/s41579-018-0029-9

b. https://www.ncbi.nlm.nih.gov/pmc/articles/PMC5861821/

c. https://bmcbiol.biomedcentral.com/articles/10.1186/s12915-014-0087-z

d. https://academic.oup.com/femsle/article/219/1/87/505487

4. Results: This section requires the most language editing and more elaboration. There is no flow in this section as the authors seemed to describe all the data analyzed. The authors should also include p-values or any other appropriate statistical measures whenever a significant/non-significant change claim is made. Moreover, readers could benefit greatly from a graphical representation of the methodology and analysis to understand what the message of the paper is and how the authors planned to answer the research questions posed.

5. Figures: The quality of all figures provided are very poor. Many of the figures are also attached separately and haphazardly, making the review process very unpleasant. All the barplot figures showcasing the differences in some bacterial phyla/genera could easily be collapsed into 1-2 figures because they carry no meaningful information in deducing changes between the phyla/genera. In truth, the final 2 figures are the only ones carrying any worthwhile information with some statistical analysis done. Moreover, it is very unprofessional for figures to just be screenshots of the software with the data visible.

6. Figure legends: All figure legends are too short and do not help in interpreting the figures.

7. Discussion: Requires more in-depth elaboration and discussion of the results.

a. “The high alpha diversity of the gut microbiota” requires a citation.

b. The authors mentioned “The genetic characteristics of different rodent strains appear to influence the diversity of the gut microbiota”, yet very little was discussed or mentioned regarding the strain identity of the animals used in the experiments.

c. Many of the interpretations derived from the results are coupled with a substantial amount speculations from specific studies that support the authors’ claims, which is not a scientifically-sound practice.

8. Conclusion: From the results and discussion, it is very difficult to reach the same conclusion the authors came to.

Minor comments:

1. Many grammatical/language erros, including:

a. Abstract: “genetically susceptibility” should be “genetic susceptibility”.

b. Introduction: The first sentence seems unnecessary. “HDACS” should be “HDACs”. “At the Pavlov Institute of Physiology of the Russian Academy…” should just be shortened into “A study…” or citing the author’s name.

Reviewer #2: The authors compared the changes of gut microbiota after stress in two rat strains that have been inbred to have low (HT) and high excitability (LT) threshold. Previous studies have evaluated their behavior and changes in inflammatory markers. In this study, the authors observed a few differences in the gut microbiota between the two strains in control conditions, such as higher diversity in the HT strain. After stress, both groups showed decrease in relative abundance of Lactobacillus. HT group showed increases in two other species. The authors concluded that the stress resistance in HT strain could be related to its higher gut microbiota diversity, but further research is needed. The authors did not draw any strong conclusions that were not supported by the data and focused more on describing the microbiota changes. The study is well designed, but the results section should contain more details for clarity. Some sentences contain grammar errors that made them hard to understand and should be edited.

Comments:

1. The results section can include a bit more details to facilitate the readers. It would be good to indicate the strain, treatment groups, and time points when describing the data in text. For example, it is not clear which time point used for the comparison for results in figure 1.

2. Fig.6 – It is not clear which time point/treatment do the data represent.

3. Fig. 7 – It would be nice to have side-by-side comparison between LT and HT strains for the genus shown, even if one group did not show significant changes. Control group data should also be included to rule out effects not related to stress.

4. Fig. 7 - Are there any changes at the genus level observed only in the LT strain?

5. Fig.3 – The color schemes are a bit hard to read. The authors could consider other color schemes that are more distinguishable between neighboring species. It would also help to make the same species the same color between graphs for comparison.

6. PLOS authors have the option to publish the peer review history of their article (what does this mean?). If published, this will include your full peer review and any attached files.

Reviewer #1: **Yes: **Andrew Octavian Sasmita

Reviewer #2: No

---

## [Author Response · Author response to Decision Letter 0]

1 Nov 2023

Thank you for your careful reading of our manuscript and valuable comments. 

Below are our answers to each of them.

Reviewer #1: The article needs a major revision and language editing before it could be considered for publication. The manuscript’s content is also rudimentary and structured very poorly. See comments below:

Major comments:

1. Introduction: The three main pathways mentioned should also include the HPA axis and the enteric nervous system.

We have added information to the introduction (lines 59-65)

2. Objective: What is the vision behind the experiments performed? What is the conceptual goal and what are the implications of this study that warrants the reader’s attention?

The goal of this study is to investigate alterations in the gut microbiota of rat strains with high and low excitability both under normal conditions and following chronic stress exposure. Given that these rat strains exhibit varying susceptibility to post-stress neuroinflammation and behavioral disturbances, understanding the precise taxonomic changes in the microbial community in response to stressors will provide insights into the influence of individual stress susceptibility on the microbiota, with broader implications for personalized medicine and interventions in stress-related disorders.

We have added information to the introduction (lines 98-104)

3. Methods: For all software used, please include their respective versions. Even with published data regarding the HT or LT strains, the authors should consider including validation of the strains for clarity of the manuscript. As this is a microbiome study, content of diet should also be mentioned. Preprocessing step is crucial and not mentioned. For reference:

a. https://www.nature.com/articles/s41579-018-0029-9

b. https://www.ncbi.nlm.nih.gov/pmc/articles/PMC5861821/

c. https://bmcbiol.biomedcentral.com/articles/10.1186/s12915-014-0087-z

d. https://academic.oup.com/femsle/article/219/1/87/505487

Thank you for your recommendation and references! In the Materials and Methods section, information regarding excitability thresholds determination has been added. In the Results section, a graph displaying the threshold values for the studied strains is included.

We have added information about diet and preprocessing.

4. Results: This section requires the most language editing and more elaboration. There is no flow in this section as the authors seemed to describe all the data analyzed. The authors should also include p-values or any other appropriate statistical measures whenever a significant/non-significant change claim is made. Moreover, readers could benefit greatly from a graphical representation of the methodology and analysis to understand what the message of the paper is and how the authors planned to answer the research questions posed.

We have completely rewritten the results section: structured the presentation of the data, provided relevant figures and statistical processing data (p values).

5. Figures: The quality of all figures provided are very poor. Many of the figures are also attached separately and haphazardly, making the review process very unpleasant. All the barplot figures showcasing the differences in some bacterial phyla/genera could easily be collapsed into 1-2 figures because they carry no meaningful information in deducing changes between the phyla/genera. In truth, the final 2 figures are the only ones carrying any worthwhile information with some statistical analysis done. Moreover, it is very unprofessional for figures to just be screenshots of the software with the data visible.

All the figures have been revised taking into account the comments.

6. Figure legends: All figure legends are too short and do not help in interpreting the figures.

The necessary information, abbreviations, and information about statistical processing have been added to the figure legends. 

7. Discussion: Requires more in-depth elaboration and discussion of the results.

a. “The high alpha diversity of the gut microbiota” requires a citation.

b. The authors mentioned “The genetic characteristics of different rodent strains appear to influence the diversity of the gut microbiota”, yet very little was discussed or mentioned regarding the strain identity of the animals used in the experiments.

c. Many of the interpretations derived from the results are coupled with a substantial amount speculations from specific studies that support the authors’ claims, which is not a scientifically-sound practice.

We have added material to the discussion in accordance with comments a-c 

8. Conclusion: From the results and discussion, it is very difficult to reach the same conclusion the authors came to.

We have added information to the conclusion section in accordance with the comments, in addition, the conclusions now are in consistent with new figures and the description of the data. 

Minor comments:

1. Many grammatical/language erros, including:

a. Abstract: “genetically susceptibility” should be “genetic susceptibility”.

b. Introduction: The first sentence seems unnecessary. “HDACS” should be “HDACs”. “At the Pavlov Institute of Physiology of the Russian Academy…” should just be shortened into “A study…” or citing the author’s name.

Fixed.

Reviewer #2: The authors compared the changes of gut microbiota after stress in two rat strains that have been inbred to have low (HT) and high excitability (LT) threshold. Previous studies have evaluated their behavior and changes in inflammatory markers. In this study, the authors observed a few differences in the gut microbiota between the two strains in control conditions, such as higher diversity in the HT strain. After stress, both groups showed decrease in relative abundance of Lactobacillus. HT group showed increases in two other species. The authors concluded that the stress resistance in HT strain could be related to its higher gut microbiota diversity, but further research is needed. The authors did not draw any strong conclusions that were not supported by the data and focused more on describing the microbiota changes. The study is well designed, but the results section should contain more details for clarity. Some sentences contain grammar errors that made them hard to understand and should be edited.

Comments:

1. The results section can include a bit more details to facilitate the readers. It would be good to indicate the strain, treatment groups, and time points when describing the data in text. For example, it is not clear which time point used for the comparison for results in figure 1.

2. Fig.6 – It is not clear which time point/treatment do the data represent.

3. Fig. 7 – It would be nice to have side-by-side comparison between LT and HT strains for the genus shown, even if one group did not show significant changes. Control group data should also be included to rule out effects not related to stress.

4. Fig. 7 - Are there any changes at the genus level observed only in the LT strain?

1 – 4: New figures have been made on which time points and comparisons are clear

5. Fig.3 – The color schemes are a bit hard to read. The authors could consider other color schemes that are more distinguishable between neighboring species. It would also help to make the same species the same color between graphs for comparison.

A new color scheme was used in Figure 6, which now reflects this data.

---

## [Decision Letter · Decision Letter 1]

14 Nov 2023

PONE-D-23-27857R1Post-stress changes in the gut microbiome composition in rats with different levels of nervous system excitabilityPLOS ONE

Dear Dr. Shalaginova,

Thank you for submitting your manuscript to PLOS ONE. After careful consideration, we feel that it has merit but does not fully meet PLOS ONE’s publication criteria as it currently stands. Therefore, we invite you to submit a revised version of the manuscript that addresses the points raised during the review process.

We look forward to receiving your revised manuscript.

Kind regards,

Brenda A Wilson, Ph.D.

Academic Editor

PLOS ONE

**Additional Editor Comments:**

While most of the concerns of the previous review have been adequately addressed, there still remain some from Reviewer 2 that need to be better addressed and some new concerns regarding the new data.

Reviewers' comments:

Reviewer's Responses to Questions

**Comments to the Author**

1. If the authors have adequately addressed your comments raised in a previous round of review and you feel that this manuscript is now acceptable for publication, you may indicate that here to bypass the “Comments to the Author” section, enter your conflict of interest statement in the “Confidential to Editor” section, and submit your "Accept" recommendation.

Reviewer #2: (No Response)

2. Is the manuscript technically sound, and do the data support the conclusions?

Reviewer #2: Partly

3. Has the statistical analysis been performed appropriately and rigorously? 

Reviewer #2: Yes

4. Have the authors made all data underlying the findings in their manuscript fully available?

Reviewer #2: Yes

5. Is the manuscript presented in an intelligible fashion and written in standard English?

Reviewer #2: Yes

6. Review Comments to the Author

Reviewer #2: The figures have been improved. However, with the new data included, I have some concerns.

1. My major concern is the difference between the “control” group time 0 and the “before stress” group. Theoretically these two groups should be similar as both groups are not treated yet. However, there are significant differences in Shannon index (Figure 3A) and levels of g. Prevotella and g. Faecalibacterium (Figure 5). The conclusion that HT strain has more diversity than LT strain is not supported if comparing using the “before stress” groups. Can the authors explain the potential differences?

2. In figure 2, combining data points from HT control group for Chao1 index is questionable as the day 24 is quite different from day 0.

7. PLOS authors have the option to publish the peer review history of their article (what does this mean?). If published, this will include your full peer review and any attached files.

Reviewer #2: No

---

## [Author Response · Author response to Decision Letter 1]

18 Nov 2023

Thank you for your careful reading of our manuscript and valuable comments. 

Below are our answers to each of them.

Reviewer #2: 

1. My major concern is the difference between the “control” group time 0 and the “before stress” group. Theoretically these two groups should be similar as both groups are not treated yet. However, there are significant differences in Shannon index (Figure 3A) 

Fig 3 - The horizontal axis on panel A shows the animals of the control groups at the corresponding time after the end of stress in the experimental groups. The horizontal axis on panel B shows the intact animals of the consolidated control. As we noted (lines 219-220) we did not find any statistically significant effect of stress on alpha diversity indexes in LT and HT rats (paired t-test).

2. and levels of g. Prevotella and g. Faecalibacterium (Figure 5). The conclusion that HT strain has more diversity than LT strain is not supported if comparing using the “before stress” groups. Can the authors explain the potential differences?

We noted (lines 245-248), that we found that in one case (Fig 5, g. Prevotella) the control HT rats significantly differed from the experimental group before stress, which makes it incorrect to compare control groups with experimental ones to assess the effect of stress in this case. But in case of g. Faecalibacterium - there were no significant differences (case/control - unpaired t-test).

We have explained the potential differences of g. Prevotella (Lines 317-324)

3. In figure 2, combining data points from HT control group for Chao1 index is questionable as the day 24 is quite different from day 0.

We specifically checked the absence of significant differences in microbiota diversity indices in rats stool of the experimental group before and at different periods after stress. As the paired t-test shows, there are no significant differences. However, the variability of the data is indeed higher in the HT group both in the case of the Ciao1 index and in the case of the Shannon index. This is also in accordance with the assumption of greater dynamism in the fluctuations of microbiota composition in the stool of high-excitable rats compared with low-excitable ones.

---

## [Decision Letter · Decision Letter 2]

27 Nov 2023

Post-stress changes in the gut microbiome composition in rats with different levels of nervous system excitability

PONE-D-23-27857R2

Dear Dr. Shalaginova,

We’re pleased to inform you that your manuscript has been judged scientifically suitable for publication and will be formally accepted for publication once it meets all outstanding technical requirements.

Kind regards,

Brenda A Wilson, Ph.D.

Academic Editor

PLOS ONE

Additional Editor Comments (optional):

Reviewers' comments:

Reviewer's Responses to Questions

**Comments to the Author**

1. If the authors have adequately addressed your comments raised in a previous round of review and you feel that this manuscript is now acceptable for publication, you may indicate that here to bypass the “Comments to the Author” section, enter your conflict of interest statement in the “Confidential to Editor” section, and submit your "Accept" recommendation.

Reviewer #2: All comments have been addressed

2. Is the manuscript technically sound, and do the data support the conclusions?

Reviewer #2: Yes

3. Has the statistical analysis been performed appropriately and rigorously? 

Reviewer #2: Yes

4. Have the authors made all data underlying the findings in their manuscript fully available?

Reviewer #2: Yes

5. Is the manuscript presented in an intelligible fashion and written in standard English?

Reviewer #2: Yes

6. Review Comments to the Author

Reviewer #2: (No Response)

7. PLOS authors have the option to publish the peer review history of their article (what does this mean?). If published, this will include your full peer review and any attached files.

Reviewer #2: No

---

## [Editor Report · Acceptance letter]

1 Dec 2023

PONE-D-23-27857R2 

*Post-stress changes in the gut microbiome composition in rats with different levels of nervous system excitability*  

Dear Dr. Shalaginova:

I'm pleased to inform you that your manuscript has been deemed suitable for publication in PLOS ONE. Congratulations! Your manuscript is now with our production department. 

Kind regards, 

on behalf of

Dr. Brenda A Wilson 

Academic Editor

PLOS ONE